



**Structural studies in active caldera geothermal systems. Reply to**
**Comment on "Estimating the depth and evolution of intrusions at**
**resurgent calderas: Los Humeros (Mexico)" by Norini and**
**Groppelli (2020).**
Stefano Urbani[1], Guido Giordano[1,2], Federico Lucci[1], Federico Rossetti[1], Valerio Acocella[1], Gerardo
Carrasco- Núñez[3]
[1]Dipartimento di Scienze, Università degli Studi Roma Tre, Largo S.L. Murialdo 1, I-00146 Roma, Italy
[2]CNR - IGAG Montelibretti, Roma
[3]Centro de Geociencias, Universidad Nacional Autónoma de México, Campus UNAM Juriquilla, 76100, Queretaro,
Mexico
Correspondence to: stefano.urbani88@gmail.com
**Abstract** Structural studies in active caldera systems are widely used in geothermal exploration to reconstruct
volcanological conceptual models. Active calderas are difficult settings to perform such studies mostly because of the
highly dynamic environment, dominated by fast accumulation of primary and secondary volcanic deposits, the variable
and transient rheology of the shallow volcanic pile, and the continuous feedback between faulting and geothermal fluid
circulation/alteration that tend to obliterate the tectonic deformation structures. In addition, deformation structures can be
also caused by near- and far-field stress regimes, which include magmatic intrusions at various depths (volumes and
rates), the evolving topography and regional tectonics. A lack of consideration of all these factors may severely underpin
the reliability of structural studies. By rebutting and providing a detailed discussion of all the points raised by the comment
of Norini and Groppelli (2020) to the Urbani et al. (2020) paper, we take the opportunity to specify the scientific rationale
of our structural fieldwork and strengthen its relevance for geothermal exploration/exploitation in active caldera
geothermal systems in general, and, particularly, for the Holocene history of deformation and geothermal circulation in
the Los Humeros caldera. At the same time, we identify several major flaws in the approach and results presented in
Norini and Groppelli (2020).
**1 Introduction**
Structural studies in active calderas provide key elements for the exploration of geothermal systems and greatly contribute
to the development of conceptual models for their exploitation. We herein reply to the comment by Norini and Groppelli
(2020) (hereafter referred to as N&G2020) on our paper Urbani et al. (2020) entitled "Estimating the depth and evolution
of intrusions at resurgent calderas: Los Humeros (Mexico)", giving us the opportunity to better discuss our approach,
results and the proposed reconstruction of the Holocene volcano-tectonic evolution of the Los Humeros Volcanic
Complex (LHVC; Mexico) and their relevance for understanding of the active geothermal system.

**2 Reply to the criticism raised in the comment**
The N&G2020's criticism on the Urbani et al. (2020) paper revolves around three main aspects: (1) supposed poor
structural field data and supposed geometric and structural inconsistency between the Holocene deformation and the
proposed subsurface model; (2) supposed lack of validation of the obtained results with those available from well-logs
data; and (3) supposed contradictions with the available stratigraphic reconstruction and radiometric ages.
Here follows a point-by-point discussion of the critical points raised in N&G2020.

*2.1 Las Papas and Las Viboras structures: inventory vs. selection method of structural analysis in active volcanic areas*



N&G2020 question the reinterpretation made by Urbani et al. (2020) of Las Papas and Las Viboras structures as presently
inactive morphological scarps, showing small-scale faults in the Cuicuiltic Member (Fig. 2 of N&G2020), and criticizing
on the supposed few data presented. In contrast to the inventory method followed by N&G2020, where all faults are
mixed together without any hierarchy and discussed as unweighted data, in Urbani et al. (2020) we followed a selection
method, with faults ranked adopting the following criteria in the field: (i) the topographic expression of the fault, (ii) the
amount of displacement of individual fault strands and/or fault systems; (iii) the along-strike persistence of the fault trace;
(iv) presence of clear kinematic indicators; (v) presence or absence of associated hydrothermal alteration, and (vi) the
relative age with respect to the Holocene intracaldera Cuicuiltic Member fall deposit; the latter being assumed as a
reference space-time marker to discriminate older or younger than 7.3 ka faults, according to its well-known age and
distribution (Dávila-Harris and Carrasco-Núñez, 2014; Carrasco-Núñez et al., 2017a). Accordingly, in Urbani et al. (2020)
we described only selected faults showing clear m-scale offsets, consistent lateral extent and evidence of hydrothermal
alteration. We therefore strongly reject that our data are poor, because they refer to the structures that, based on the above
listed criteria, allowed us to discriminate and rank volcano-tectonic and hydrothermal processes, which are instead missed
by the inventory method of unweighted fault dataset adopted by N&G2020. In terms of geothermal exploration, the faults
presented in N&G2020 are indeed questionable in terms of relevance. For example, the LH17106 and LH62 outcrops
shown in Fig. 2c of N&G2020 are in the same location of outcrop LH-08 shown in Fig. 5c of Urbani et al. (2020), where
an erosional surface at the top of undeformed and unaltered pyroclastic deposits of the Xoxoctic Tuff, blanketed by the
Cuicuiltic Member is clearly visible. Fig. 1a shows the same outcrop, where the erosional unconformity at the top of the
underlying subhorizontal pyroclastics is sutured by the Cuicuiltic Member fall deposits. The large-scale blanketing
geometry of the unaltered Cuicuiltic Member fall deposits across the Las Papas scarp is well visible in Fig. 1b. This
indicates that Las Papas is currently an inactive morphological scarp without evidence of hydrothermal alteration.
Whether or not this scarp was associated in origin (prior to 7.3 ka) with a fault scarp is not evident in the field nor relevant
for our study, focused on present-day relationships between faulting and geothermal circulation. Noteworthy, even Norini
et al. (2019; see sections 4 and 6.2) raise doubts on the relevance of the Las Papas structure within the Los Humeros
geothermal field, suggesting a weak or no connection with the geothermal reservoir. The same holds for the Las Viboras
structure.
In our opinion, the small-scale faults shown by N&G2020 in their Fig. 2d-e are not at all compelling and may be
alternatively interpreted as small-scale normal faults generated by near-field (local) stresses affecting unlithified material
(e.g., Wernicke and Birchfiel, 1982; Bridgewater et al., 1985; Branney and Kokelaar, 1994; Gao et al., 2020; Yang and
Van Loon, 2016). In particular, Fig. 2d of N&G2020 is unclear, whereas their Fig. 2e does not even show any
displacement of the lower white and black pumice beds, as well as on the upper brown beds, suggesting an
intraformational readjustment (Fig. 1c) rather than a fault. (e.g. Van Loon and Wiggers, 1975, 1976; Branney and
Kokelaar, 1994). N&G2020 fail to discuss any possible alternative origin for their small-scale faults, which, considered
the location in an active caldera floor, severely impinges the reliability of the inventory dataset presented in N&G2020
and its relevance for geothermal studies. Reinterpreting the small-scale offsets shown in Fig. 2 on N&G2020 as minor
gravitational structures (i) would justify why they have no connection with the geothermal circulation nor with any
thermal anomaly, and (ii) clarifies to the reader why the Urbani et al (2020) paper instead focused only on faults that were
ranked as of first-order importance in terms of displacement, persistence in the field and age of the structurally-controlled
fluid circulation.
In summary, we (i) question the use of the inventory method for structural analysis adopted by N&G2020 when applied
to active calderas, which may prove to be inappropriate and unable to discriminate between first-order, deep and





geothermally relevant, fault systems from small-scale, soft-state deformation structures that are also common in intracaldera domains (e.g., Branney and Kokelaar, 1994); and (ii) consider the selection method of structural analysis used by Urbani et al. (2020) as fully appropriate to rank deformation structures (Fig. 1a-i) when the aim of a structural fieldwork is to characterize the surface deformation related to the recent activity of a caldera, to constrain the morphotectonic fingerprints of the resurgence, to evaluate its source and areal extent and, even more importantly, its relevance for the active geothermal system.

### 2.2 Arroyo Grande and Maxtaloya faults: the importance of tracking fluid path migration in space and time

N&G2020 state that *"active/fossil alteration doesn't always allow identifying faults or the age of faulting, because it depends also on their depth, life span of the hydrothermal system, spatial relationships, and fluid paths along primary permeability and faults zones (Bonali et al., 2016; Giordano et al., 2016)"*. The two studies cited by N&G2020 are in no way at odds with Urbani et al. (2020). The work of Bonali et al. (2016), on the active tectonics at Copahue (Argentina) points out that active fault systems in volcanic settings are responsible for driving hot fluids to the surface. Similarly, the works of Giordano et al. (2013; 2016) on the Tocomar geothermal field (Puna Plateau, Argentina), investigated the evidence of a geothermal field based of the overlapping distribution of hot springs and active fault systems. We thank N&G2020 for reporting to our attention these two very interesting papers because, along with mainstream literature, they clearly indicate that hydrothermal fluids and associated alteration in volcanic settings are driven/controlled by active fault systems. The relationship between faulting and fluid circulation is well established also in exhumed systems, where it is clear how fault-permeability is affected by the interplay between far-field regional stress field and the near-field stress regime (e.g. Rossetti et al. 2011; Olvera Garcia et al. 2020). Therefore, the cited papers support the proposal of Urbani et al. (2020) to use the distribution and intensity of the hydrothermal alteration within the 7.3 ka Cuicuiltic Member marker beds, that ubiquitously blanket the caldera floor and all the fault scarps, as a valid space-time marker in the field to discriminate active vs. inactive fault segments controlling the upwelling of geothermal fluids (Fig. 1d-i). Concluding, contrary to N&G2020, we reaffirm that, in agreement with authoritative literature, hydrothermal alteration follows the space-time distribution of structurally-controlled (fault-induced) secondary permeability pathways and its distribution should be used, along with measured fault displacements, persistence and (relative) age, as an indication of fault activity and ranking for geothermal purposes. At Los Potreros, the presence of the 7.3 ka Cuicuiltic Member marker bed allows to track the type and intensity of deformation and its association with fluid circulation and alteration in space and time.

### 2.3 Surface thermal anomalies

N&G2020 state *"The Maxtaloya fault trace is coincident with a sharp thermal anomaly identified by Norini et al. (2015). Urbani et al. (2020) didn't consider this positive (warm) anomaly when they discussed the thermal remote sensing results published by Norini et al. (2015) (Section 5.3 in Urbani et al., 2020)"*. This statement is not correct, as clearly written in section 5.3 of Urbani et al. (2020). Moreover, the sharp and narrow temperature peaks, spatially coincident with the Los Humeros and Loma Blanca faults described by Urbani et al. (2020), are further supported by the recent work of Jentsch et al. (2020; also presented in Deliverable 4.3 of GEMex, 2019a), where soil temperature anomalies (T > 43°C) are identified only at Los Humeros and at Loma Blanca areas, whereas no thermal anomaly is recognized along other sections of the Maxtaloya fault (see Fig. 5a in Jentsch et al. 2020). We therefore reject the criticism from N&G2020, who instead failed to consider the recent results presented by Jentsch et al. (2020).

### 2.4 Identification and geometry of uplifted areas: topographic data and structural mapping



N&G2020 criticize the location and geometry of the three uplifted areas of Los Humeros, Loma Blanca and Arroyo
Grande identified by Urbani et al. (2020). However, in the topographic profiles across the bulges shown by N&G2020 in
their Fig. 4a-b, the uplifted areas at Loma Blanca, Arroyo Grande and Los Humeros are well visible and their existence
is unquestionable. Therefore, it is unclear on what basis N&G2020 question the existence of such uplifted areas. The
asymmetry (Arroyo Grande) and tilt of the uplifted areas (Loma Blanca) detailed by N&G2020 are in no way adversative
to the Urbani et al (2020) interpretation. Again, it is unclear why these shapes are reported as counterproofs. Asymmetric
bulges are common characteristics in many volcanic regions worldwide, in resurgent calderas (e.g. Ischia, Pantelleria,
Sierra Negra and Alcedo; Galetto et al. 2017 and references therein) or associated with shallow intrusions, such as Usu
(Goto and Tomiya, 2019), Chaine de Puys (van Wyk de Vries et al., 2014; Petronis et al 2019), Bezymianny (Gorshkov,
1959) and Mt St. Helens (Lipman, et al. 1981). Despite being stimulating for future works, investigation of the exact
origin of the bulge shapes was far beyond the scope of Urbani et al. (2020), who, for this reason, maintained the same
initial and simplified geometric configuration for their analogue models. Therefore, the comment made by N&G2020 is
not relevant for the discussion presented in Urbani et al. (2020).
*2.4.1 Apical depression of bulges*
The model proposed by Urbani et al. (2020) predicts the formation of an apical depression on the top of a bulge induced
by a shallow intrusion. N&G 2020 state that the topography of natural bulges identified by Urbani et al. (2020) does not
show well-defined apical depressions in the asymmetric Arroyo Grande and Los Humeros uplifted areas, contradicting
the model results. Analogue modeling in Urbani et al (2020) inject symmetric intrusions, a condition appropriate for the
morphology of the Loma Blanca bulge, where the apical depression is very well evident (Fig. 2) and measured in the field
(Fig. 6f in Urbani et al. 2020). The Arroyo Grande and Los Humeros bulges are instead asymmetrical, and likely
developed as trapdoor uplifts (thus without apical depression) associated with asymmetric intrusions and with a
deformation amount much larger than that at Loma Blanca and that considered in the analogue models. Therefore, the
comment made by N&G2020 is incorrect regarding the Loma Blanca bulge and not relevant in the other two cases,
therefore not compromising in any way the predictive value of the model proposed in Urbani et al. (2020).
*2.4.3 Reverse faults bounding uplifted areas*
N&G2020 state that Urbani et al. (2020) do not provide independent validation of their multiple magmatic intrusion
model, such as field evidence of reverse faults predicted by the analogue modeling results. Exposure of faults in active
caldera floors depends on many factors: (i) elastic versus anelastic response to deformation source, its location, intensity
and duration, (ii) nucleation depth and propagation up to surface, (iii) rate of burial versus exhumation rates. Therefore,
while reverse faults accompanying both large-scale resurgence and local uplifts are expected by any model, the scarcity
of visible and measurable reverse faulting in no way disproves the intrusion of cryptodomes and resurgence (Bonanza,
Lipman et al., 2015; Long Valley, Hildreth et al., 2017; Kutcharo, Goto and McPhie, 2019). Therefore, the statement by
N&G2020 claiming that the locations of such reverse faults "*are a fundamental feature of their model*" is incorrect. In
addition, N&G2020 show the traces of inferred reverse faults at the periphery of the Loma Blanca bulge, just where the
Urbani et al. 2020 model predicts (see Fig. 2), making their own statements really unclear.
**2.5 Validation of the proposed model: geothermal wells log data**
*2.5.1 Lithology of intrusions*
N&G2020 claim the lack of validation of the models proposed in Urbani et al. (2020) also invoking the thermal profile
and the stratigraphy of the H4 well drilled on the top of the Loma Blanca bulge. First, we would like to emphasize that




the proposed reinterpretation of the subsurface stratigraphy presented in Urbani et al. (2020) is not just based on the H4
well. A great part of section 2 ("Geological-structural setting") and Figs. 2a-b presented in Urbani et al. (2020) discuss in
detail the published data from twelve well logs (including the H4 well log) as presented in Arellano et al. (2003) and in
Carrasco-Núñez et al. (2017a, 2017b). The model evaluation of the intrusion depths, as derived from the equation of
Brothelande and Merle (2015), are valid within the modelling assumptions and are within the depth range of some
rhyolitic-dacitic bodies drilled in geothermal wells, wherein they are simply described texturally as lavas (Carrasco-Núñez
et al., 2017b and references therein). The lithologic definition of "lava" is associated with aphanitic to phaneritic textures
that are not only restricted to subaerial environments and may be impossible to distinguish from textures of sub-
volcanic/hypabyssal bodies. Hypabyssal rocks are characterized by a rapid cooling and their textures are fine grained or
glassy, and mostly resemble those of volcanic rocks (Phillpots and Ague, 2009). One of the most famous examples of
felsic hypabyssal intrusions in intracaldera ignimbrite deposits is in Long Valley Caldera (California). At Long Valley,
the well logs revealed ca. 300 m cumulative thick succession of aphanitic to phyric rhyolitic intrusions emplaced during
the post-caldera stage, into the older, ca. 1200 m thick, intracaldera Bishop Tuff (McConnell et al., 1995). We therefore
reject the criticism by N&G2020 only based on uncritical reading of published well-log litho-stratigraphies.
*2.5.2 Geometry of caldera fill*
The reinterpretation proposed by Urbani et al. (2020) of some of the rhyolitic-dacitic bodies of the Los Potreros subsurface
as hypabyssal intrusives is not simply based on their lithology, but also on their geometry, stratigraphic position, as well
as the whole geometry of the caldera fill; all elements neither considered nor discussed in N&G2020. When correlating
the stratigraphic well-logs, Urbani et al. (2020) documented (in section 2 at p. 530 and Fig. 2) the irregular geometry of
both the top of the Xaltipan intracaldera ignimbrite and the post-caldera units, as well as the lack of a clear topography
filling geometry: a stratigraphic setting that can be hardly reconciled with an intracaldera setting unless the emplacement
of intrusive bodies has occurred in the shallow crust. Noteworthy, the main geometric anomalies of the caldera fill appear
right in correspondence with the possible location of a felsic intrusion. For example, a 600 m-thick rhyolitic-dacitic body
showing all the petrographic features of a hypabyssal intrusion is reported to the west of Arroyo Grande in the H20 well
at 470-1060 m of depth from the surface (see also Carrasco-Núñez et al. (2017b). It is located at the top of the pre-caldera
andesites, intrudes both the intracaldera and the post-caldera units, and shows no lateral continuity. Similar felsic bodies
were also drilled in H5, H26, H19 and H25 wells. Furthermore, N&G2020 completely misinterpreted and misquoted a
recent work by Cavazos-Alvarez et al. (2020), which only deals with the reinterpretation of andesitic layers within the
Xaltipan intracaldera ignimbrite (see blue ellipses in wells: H10-Fig. 3a; H20-Fig. 3b; and H42-Fig. 3e) and does not
question the interpretation of the rhyolite bodies proposed by Urbani et al., 2020 as small intrusions located above and
below the Xaltipan ignimbrite. With regard to these rhyolite bodies, Cavazos-Alvarez et al. (2020) not only confirm their
existence in wells H20 and H26 (red ellipses in Figs. 3b and 3d), but also identify previously unrecognized (i) ca. 400 m
cumulative thick rhyolite layers (between ca. 500-1000 m below the surface) in well H25 (Fig. 3c), and (ii) a ca. 50 m
thick rhyolite layers (between 850-900 m below the surface) in well H42 (Fig. 3e). The depths of these rhyolitic layers
are compatible with the estimated intrusion depth of 425 ±170 m proposed by Urbani et al. (2020) for the emplacement
of small cryptodomes within the volcanic sequence. It should be emphasized that the presence of rhyolitic bodies within
the volcanic sequence in the Los Potreros intracaldera domain is also reported in the geological cross-section included in
the recently updated geological map of Los Humeros (Carrasco-Núñez et al. 2017a). Summarizing, we have demonstrated
the agreement between the works of Carrasco-Núñez et al. (2017a, 2017b), Urbani et al. (2020) and Cavazos-Alvarez et





al. (2020) for what concerns the subsurface stratigraphy of the Los Potreros intracaldera domain, and therefore we reject
the criticism of N&G2020.

### 2.5.3 Thermal gradient

The statement by N&G2020 on the absence of an in-depth sharp increase of the temperature and geothermal gradient in
the H4 well (considered to remain constant at ca. 20 °C/km; see Fig. 3d in N&G2020) is not correct. The existing published
in-depth temperature profiles of the H4 well (Fig. 4a; after Torres-Rodriguez, 1995; Prol-Ledesma, 1988, 1998; Martinez-
Serrano, 2002) show a clear sharp temperature increase (+150 °C) in less than 200 m, up to 300 °C at 1000 m below the
surface. The temperature profile is then characterized by a progressive temperature decrease down to ca. 200 °C at 2000
m depth. Such temperature profile is not observed in the very close H43 well (Fig. 4a, after Lorenzo-Pulido, 2008).
Significantly, on the top of the Loma Blanca bulge, very close to the H4 well, Norini et al. (2019) and also N&G2020
report *"a warm normal fault"* in the Cuicuiltic Member deposits and documented it through a thermal image (Fig. 5b in
Norini et al., 2019; Fig. 3 in N&G2020, Figs. 1e-g in this reply), confirming the active thermal activity in the Loma
Blanca area. Furthermore, 300 m away from the H4 well, at the southern termination of the Loma Blanca fault, Jentsch
et al. (2020) measured the highest surface temperature (91.3 °C) of the whole Los Potreros caldera, corresponding to an
active solfatara (Figs. 1h-I, 4b).

### 2.6 Validation of the proposed model: stratigraphic and radiometric data

### 2.6.1 Age of the domes along Los Humeros fault

N&G2020 question the presence of domes younger than 7.3 ka based on stratigraphic and radiometric data presented in
Carrasco-Núñez et al. (2018). Fig. 5 shows, in agreement with the geological map of Carrasco-Núñez et al. (2017a), a
perspective view of the Los Potreros caldera floor across the Maxtaloya and Los Humeros faults. The images show the
presence of lava domes and flows of variable composition both covered by and emplaced above the 7.3 ka Cuicuiltic
Member. Older lavas include those associated with the "Resurgent Phase" (50.7-44.8 ka; "Qr1") in Carrasco-Núñez et
al., (2018 and references therein). Younger lavas show absence of the 7.3 ka Cuicuiltic member cover and a morphology
poorly or unaffected by evidence of faulting. We therefore conclude that field evidence supports Urbani et al 2020 in
documenting the presence of lava bodies younger than 7.3 ka issued along the Maxtaloya-Los Humeros faults.

### 2.6.2 Recent history of caldera floor uplift

N&G2020 misquote Urbani et al. (2020), attributing to them the interpretation of a northward shift in volcanic activity
within the Los Potreros caldera, which was neither declared nor intended in the paper. Urbani et al (2020) simply
summarize field evidence stating *"the recent (post-caldera collapse) uplift in the Los Potreros caldera moved*
*progressively northwards, from the south and north-eastern sector of the caldera towards the north along the Los*
*Humeros and Loma Blanca scarps"*. Urbani et al. (2020) did not discuss the causes of such northward shift and even less
attributed it to a shift in "the volcanic feeding system" as erroneously and unjustifiably reported by N&G2020. The fate
of a magma intrusion, i.e. whether it will erupt or stop in the crust, depends on many factors, such as its buoyancy (density
contrast with host rocks), the initial gas content, the rise speed and style of decompression-degassing, the rheology of the
magma and of the intruded crust, including its layering, structure and so forth. The evolution over time and space of
intrusions in a caldera may see different phases and have many different causes, partly depending on feedbacks existing
between the evolving configuration of the magmatic plumbing system and the evolving rheology and structure of the
caldera roof rocks. At Los Humeros the plumbing system of the last 10 ka has been reconstructed in detail by Lucci et al
(2020). This study documents a multistorey magmatic complex, which allows the eruption along the Los Potreros caldera



floor of both deep-sourced (>30 km) olivine basalts and shallow-differentiated (< 3 km) felsic trachytes and rhyolites.
The results of Lucci et al. (2020), curiously neither cited nor discussed by N&G2020, highlight the absence of the classic
large volume, single magma chamber and suggest that the activation of magma sources at different depths appear not to
have followed any specific pattern during the Holocene. A corollary of the present absence below Los Humeros of a
single large magma chamber/crystal mush able to form a rheological barrier to the rise of basalts directly from lower
crustal depths severely impinges upon the model of classic resurgence supported by N&G2020, which requires the
existence of a voluminous viscous layer accommodating magma recharge and acting as a pressure source for resurgence
(Galetto et al. 2017).
**3. Summary and implications for the Los Humeros geothermal system**
Understanding the anatomy of magma plumbing systems of active volcanic systems, from deeper reservoirs to subsurface
ephemeral batches, is crucial to define temperature, depth and geometry of the heat sources for geothermal exploration.
The Pleistocene-Holocene Los Humeros Volcanic Complex (LMVC, located in the eastern Trans-Mexican Volcanic Belt
(central Mexico), represents one of the most important exploited geothermal fields in Mexico, with ca. 95 MW of
produced electricity. Geological investigations at LMVC started at the end of the '70 of the last century and culminated
with the production of (i) the first comprehensive geological map (Fig. 6a; after Ferriz and Mahood (1984), (ii) a structural
map of the intracaldera domain (Fig. 6b after Alcantara et al., 1988), (iii) the proposal of a petrological conceptual model
of the plumbing system made of a single voluminous (ca. 1200 km$^3$) melt-dominated and zoned magma chamber at
shallow depths (ca. 5 km, Fig. 6c, after Verma, 1985), and (iv) the proposal of an inflation-deflation caldera
episodic/cyclic model (Fig. 6d, after Campos-Enriquez and Arredondo-Fragoso, 1992) connected to the activity of the
single voluminous conventional magma chamber of Verma (1985). Since these main studies, up to the most recent
published works, the understanding of the Los Humeros volcanic complex has been incremental, never questioning the
consolidated model of the single zoned magma chamber where all petrologic, volcanologic and deformation processes
originate (i.e., Ferriz and Mahood, 1984; Alcantara et al., 1988; Verma, 1985; Campos-Enriquez and Arredondo-Fragoso,
1992). Structural work by Norini et al. (2015, 2019), produced updates and refined versions (Figs. 6e and 6f) of the
original structural map by Alcantara et al. (1988). Based on the assumption of the existence of an active single voluminous
magma chamber as proposed in the early '1980s (Verma, 1985), post-caldera deformation has been interpreted uniquely
as due to a classic mechanism of resurgence (e.g., Fig. 6g after Norini et al., 2019) that much (or completely) resemble
the first proposal of Campos-Enriquez and Arredondo-Fragoso (1992). However, such conceptual model is now under
stress as the geothermal anomalies appear very localized, mainly confined along the NNW-SSE-trending "Maxtaloya-
Los Humeros-Loma Blanca-Los Conejos" corridor and corresponding to the almost unique, narrow thermal anomaly
recognized within the Los Potreros caldera (Norini et al., 2015; Peiffer et al., 2019; Jentsch et al., 2020), rapidly declining
away. This geothermal configuration is reflected in the low number of productive geothermal wells (ca. 25 out of ca. 60;
Gutierrez-Negrín et al. 2019; 2020) but is difficult to reconcile with the existence of a single, deep seated, large volume
magmatic source that should instead generate widespread and sustained thermal anomalies in the caldera floor, such as in
active resurgent calderas like Ischia (Carlino et al., 2014).
A step-change of paradigm in the reconstruction of the Holocene magmatic plumbing system at Los Humeros has been
proposed in Lucci et al. (2020) (not cited by N&G2020) and GEMex (2019c), with important implications for the
understanding of the present-day geothermal system. Lucci et al. (2020) carried out a thermobarometric study of all
exposed Holocene lavas, demonstrating that the scattered intracaldera monogenetic activity reflects the ascent of magmas
from basaltic to trachytic in composition from sources located at depths comprised between > 30 km (basalts) to < 3 km



(trachytes), and for variably evolved compositions, with complex histories of ascent and stalling at various depths,
depicting a multistorey plumbing system (e.g. Cashman and Giordano, 2014; Cashman et al. 2017; Sparks et al. 2019).
This innovative reconstruction of the plumbing system suggests that the large volume magma chamber at 5 km depth that
produced the caldera collapses at the time of the eruption of the Xaltipan ignimbrite (164 ka) and Zaragoza ignimbrite
(69 ka) does not exist anymore as a single melt-dominated volume, allowing the rise to surface of mantle magmas as well
as differentiation at various depths of small batches of magma through the entire crust. Urbani et al. (2020) performed a
structural fieldwork based on a selective method approach combined with analogue models, showing that, at least during
the Holocene, the classic resurgence model (e.g. Norini et al. 2019) does not explain the fault-ranks and the spatio-
temporal evolution of the deformation/alteration. This change of paradigm at Los Humeros implies: (i) the inadequacy of
the hypothesis of a single, large and voluminous shallow magmatic chamber homogeneously distributed beneath the
caldera; (ii) the proposal of an innovative scenario, characterized by a complex magmatic plumbing system vertically
distributed across the entire crust, from a deeper residence zone for basalts to a shallower magmatic plexus made of small
single-charge ephemeral pockets of heterogeneous magmas localized beneath the Los Humeros nested caldera (Fig. 7a,
after Lucci et al., 2020), and (iii) the interpretation of the recent deformation at Los Humeros volcanic complex not as a
classical resurgence associated with the bulk inflation of a deep magma reservoir, but as the response to the ascent and
emplacement of multiple, small-volume magma batches at shallow crustal conditions (< 1km depth) (Fig. 7b, after Urbani
et al., 2020). These results bear important consequences on the geothermal exploration/exploitation and siting of future
geothermal wells, where shallow magma bodies can act as scattered and localized short-lived heat sources complicating
the pattern of isotherms related to deeper reservoirs. At the same time, the evidence of absence during the Holocene of an
actively recharged large and melt-dominated magma chamber located at 5 km depth (i.e. the Xaltipan/Zaragoza magma
chamber) may help understanding the localized nature of the thermal anomaly at Los Humeros.
We are aware that our studies are valid within the framework of the data available and assumptions made, and that further
investigations in the Los Humeros caldera are necessary to confirm both the descriptive/predictive ability and limits of
our proposed models. However, we not only reject the hard judgments expressed by N&G2020 on Urbani et al. (2020),
but also think to have shown the many methodological and logical flaws in the scientific rationale followed by N&G2020.
In conclusion, while we thank N&G2020 for having given us the opportunity to better express our thoughts and defend
our model, we would also like to underline that it would have been less surprising and much more appropriate to discuss
this matter in any of the many conferences and workshops made available within the framework of our common three-
years long GEMEX Project, including the co-authoring of the D3.2 final report (GEMEX, 2019c, with full reference
therein to Urbani et al. 2020 contents and results). This would have offered the opportunity to us and to the entire GEMEX
community to make further progresses based on an open and public discussion of controversial issues rather than giving
a (misleading) formal impression, after its ending, that this important Project has taken uncertain paths.

**Data availability**
All the data presented in this paper are available upon request.

**Author contributions**
All the authors contributed equally to the preparation of this reply.

**Competing interests**
The authors declare that they have no conflict of interest.

**Acknowledgements**

The original research (Urbani et al. 2020) was funded the European Union Horizon 2020 GEMex project (grant agreement
no. 727550) and by the Mexican Energy Sustainability fund CONACYT-SENER, WP 4.5 of the project 2015-04-268074.



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

Preliminary 3-D geological models of Los Humeros and Acoculco geothermal fields (Mexico) – H2020 GEMex Project,
Adv. Geosci., 45, 321-333, https://doi.org/10.5194/adgeo-45-321-2018, 2018.
Calcagno, P., Evanno, G., Trumpy, E., Gutiérrez-Negrín, L. C., Norini, G., Macías, J. L., Carrasco-Núñez, G., Liotta, D.,
Garduño-Monroy, V. H., Páll Hersir, G., Vaessen, L., and Evanno, G.: Updating the 3D Geomodels of Los Humeros and
Acoculco Geothermal Systems (Mexico) – H2020 GEMex Project Proceedings, World Geothermal Congress 2020,
Reykjavik, Iceland, April 26 – May 2, 2020, p.12, 2020.
Campos-Enriquez, J. O., and Arredondo-Fragoso, J. J.: Gravity study of Los Humeros caldera complex, Mexico: Structure
and associated geothermal system. J. Volcanol. Geoth. Res., 49(1-2), 69-90, 1992.
Carlino, S., Somma, R., Troiano, A., Di Giuseppe, M. G., Troise, C., and De Natale, G.: The geothermal system of Ischia
Island (southern Italy): critical review and sustainability analysis of geothermal resource for electricity generation, Renew.
Energ., 62, 177-196, https://doi.org/10.1016/j.renene.2013.06.052, 2014.
Carrasco-Núñez, G., Hernández, J., De León, L., Dávila, P., Norini, G., Bernal, J. P., Jicha, B., Navarro, M., and López-
Quiroz, P.: Geologic Map of Los Humeros volcanic complex and geothermal field, eastern Trans-Mexican Volcanic
Belt/Mapa geológico del complejo volcánico Los Humeros y campo geotérmico, sector oriental del Cinturón Volcánico
Trans-Mexicano, Terradigitalis, 1, 1–11, https://doi.org/10.22201/igg.terradigitalis.2017.2.24.78, 2017a.
Carrasco-Núñez, G., López-Martínez, M., Hernández, J., and Vargas, V.: Subsurface stratigraphy and its correlation with
the surficial geology at Los Humeros geothermal field, eastern Trans-Mexican Volcanic Belt, Geothermics, 67, 1–17,
https://doi.org/10.1016/j.geothermics.2017.01.001, 2017b.
Carrasco-Núñez, G., Bernal, J. P., Davila, P., Jicha, B., Giordano, G., and Hernández, J.: Reappraisal of Los Humeros
volcanic complex by new U/Th zircon and 40Ar/39Ar dating: Implications for greater geothermal potential, Geochem.
Geophy. Geosy., 19, 132–149, https://doi.org/10.1002/2017GC007044, 2018.



Cashman, K. V. and Giordano, G.: Calderas and magma reservoirs, J. Volcanol. Geoth. Res., 288, 28–45,
https://doi.org/10.1016/j.jvolgeores.2014.09.007, 2014.
Cashman, K. V., Sparks, R. S. J., and Blundy, J. D.: Vertically extensive and unstable magmatic systems: a unified view
of igneous processes, Science, 355(6331), https://doi.org/10.1126/science.aag3055, 2017.
Cavazos-Álvarez, J. A., and Carrasco-Núñez, G.: Anatomy of the Xáltipan ignimbrite at Los Humeros Volcanic Complex;
the largest eruption of the Trans-Mexican Volcanic Belt: J. Volcanol. Geoth. Res., 392, 106755.
https://doi.org/10.1016/j.jvolgeores.2019.106755, 2020.
Cavazos-Álvarez, J. A., Carrasco-Núñez, G., Dávila-Harris, P., Peña, D., Jáquez, A., and Arteaga, D.: Facies variations
and permeability of ignimbrites in active geothermal systems; case study of the Xáltipan ignimbrite at Los Humeros
Volcanic Complex. J. S. Am. Earth Sci., 104, 102810. https://doi.org/10.1016/j.jsames.2020.102810, 2020.
Cedillo, F.: Geologia del subsuelo del campo geotermico de Los Humeros, Puebla. Internal Report HU/RE/03/97,
Comision Federal de Electricidad, Gerencia de Proyectos Geotermoelectricos, Residencia Los Humeros, Puebla, 30 pp.,
370  1997.

Charlier, B., and Zellmer, G.: Some remarks on U–Th mineral ages from igneous rocks with prolonged crystallisation
histories. Earth Planet. Sc. Lett., 183(3-4), 457-469, https://doi.org/10.1016/S0012-821X(00)00298-3, 2000.
Dávila-Harris, P. and Carrasco-Núñez, G.: An unusual syn-eruptive bimodal eruption: The Holocene Cuicuiltic Member
at Los Humeros caldera, Mexico, J. Volcanol. Geoth. Res., 271, 24–42, https://doi.org/10.1016/j.jvolgeores.2013.11.020,
375  2014.

DeRita, D., Giordano, G., and Cecili, A.: A model for submarine rhyolite dome growth: Ponza Island (central Italy). J.
Volcanol. Geoth. Res., 107(4), 221-239, 2001.
de Saint-Blanquat, M., Habert, G., Horsman, E., Morgan, S.S., Tikoff, B., Launeau, P., Gleizes, G.: Mechanisms and
duration of non tectonically assisted magma emplacement in the upper crust: the Black Mesa pluton, Henry Mountains,
Utah. Tectonophysics, 428, 1–31, https://doi.org/10.1016/j.tecto.2006.07.014, 2006.
de Vries, B. V. W., Marquez, A., Herrera, R., Bruña, J. G., Llanes, P., and Delcamp, A.: Craters of elevation revisited:
forced-folds, bulging and uplift of volcanoes. Bull. Volcanol. 76, 875, https://doi.org/10.1007/s00445-014-0875-x, 2014.
Donnadieu, F., and Merle, O.: Experiments on the indentation process during cryptodome intrusions: new insights into
Mount St. Helens deformation. Geology, 26, 79–82, https://doi.org/10.1130/0091-
7613(1998)026<0079:EOTIPD>2.3.CO;2 , 1998.
Donnadieu, F., and Merle, O.: Geometrical constraints of the 1980 Mount St. Helens intrusion from analogue models.
Geophys. Res. Lett., 28(4), 639-642, https://doi.org/10.1029/2000GL011869, 2001.
Ferriz, H., and Mahood, G. A.: Eruption rates and compositional trends at Los Humeros volcanic center, Puebla, Mexico.
J. Geophys. Res.-Sol. Ea., 89, 8511-8524, 1984.
Galland, O.: Experimental modelling of ground deformation associated with shallow magma intrusions, Earth Planet. Sc.
Lett., 317-318,145-156, https://doi.org/10.1016/j.epsl.2011.10.017, 2012.
Gao, Y., Jiang, Z., Best, J. L., and Zhang, J.: Soft-sediment deformation structures as indicators of tectono-volcanic
activity during evolution of a lacustrine basin: A case study from the Upper Triassic Ordos Basin, China. Mar. Petrol.
Geol., 115, 104250, https://doi.org/10.1016/j.marpetgeo.2020.104250, 2020.
GEMex: Final Report on geochemical characterization and origin of cold and thermal fluids. Deliverable 4.3. GEMex
project technical report, Horizon 2020, European Union, 213 pp. http://www.gemex-h2020.eu, 2019a.
GEMex: Final report on active systems: Los Humeros and Acoculco. Deliverable 4.1. GEMex project technical report,
Horizon 2020, European Union, 334 pp. http://www.gemex-h2020.eu, 2019b.



GEMex: Report on the volcanological conceptual models of Los Humeros and AcoculcoLos Humeros and Acoculco.
Deliverable 3.2. GEMex project technical report, Horizon 2020, European Union, 169 pp. http://www.gemex-h2020.eu,
2019c.
Giordano, G., and Cas, R. A.: Structure of the Upper Devonian Boyd Volcanic Complex, south coast New South Wales:
Implications for the Devonian-Carboniferous evolution of the Lachlan Fold Belt. Aust. J. Earth Sci., 48(1), 49-61, 2001.
Giordano, G., Pinton, A., Cianfarra, P., Baez, W., Chiodi, A., Viramonte, J., Norini, G., and Groppelli, G.: Structural
control on geothermal circulation in the Cerro Tuzgle–Tocomar geothermal volcanic area (Puna plateau, Argentina). J.
Volcanol. Geoth. Res., 249, 77-94. https://doi.org/10.1016/j.jvolgeores.2012.09.009, 2013
Giordano, G., Ahumada, M. F., Aldega, L., Baez, W. A., Becchio, R. A., Bigi, S., Caricchi C., Chiodi A., Corrado S., De
Benedetti A., Favetto A., Filipovich R., Fusari A., Groppelli G., Invernizzi C., Maffucci R., Norini G., Pinton A.,
Pomposiello C., Tassi F., Taviani S., Viramonte J.: Preliminary data on the structure and potential of the Tocomar
geothermal        field        (Puna        plateau,        Argentina).        Energy.        Proced.,        97,        202-209,
https://dx.doi.org/10.1016/j.egypro.2016.10.055, 2016.
Gorshkov,        G.S.:        Gigantic        eruption        of        the        volcano        Bezymianny.        Bull.        Volcanol.,        20:77–109,
https://doi.org/10.1007/BF02596572, 1959.
Goto, Y. and McPhie, J.: Tectonics, structure, and resurgence of the largest Quaternary caldera in Japan: Kutcharo,
Hokkaido, Geol. Soc. Am. Bull., 130, 1307−1322, https://doi.org/10.1130/B31900.1, 2018.
Goto, Y., and Tomiya, A.: Internal Structures and Growth Style of a Quaternary Subaerial Rhyodacite Cryptodome at
Ogariyama, Usu Volcano, Hokkaido, Japan. Front. Earth Sci. 7:66, https://doi.org/10.3389/feart.2019.00066, 2019.
Guldstrand, F., Burchardt, S., Hallot, E., and Galland, O.: Dynamics of surface deformation induced by dikes and cone
sheets    in    a    cohesive    coulomb    brittle    crust.    J.    Geophys.    Res.    Solid    Earth,    122,    8511–8524.
https://doi.org/10.1002/2017JB014346, 2017.
Guldstrand, F., Galland, O., Hallot, E., and Burchardt, S.: Experimental constraints on forecasting the location of volcanic
eruptions from preeruptive surface deformation. Front. Earth Sci. 6, 1–9. https://doi.org/10.3389/feart.2018.00007, 2018.
Gutiérrez-Negrín L.C.A.: Current status of geothermal-electric production in Mexico IOP Conf. Ser., Earth Environ. Sci.

424   249 012017. 2019.

Gutiérrez-Negrín, L.C.A. Canchola I., Romo-Jones, J.M. and Quijano-LeónJ.L.: Geothermal energy in Mexico: update
and perspectives, Proceedings World Geothermal Congress 2020 Reykjavik, Iceland, April 26 – May 2, 2020.
Hanson, R. E., and Wilson, T. J.: Large-scale rhyolite peperites (Jurassic, southern Chile), J. Volcanol. Geoth. Res., 54(3-
4), 247-264, https://doi.org/10.1016/0377-0273(93)90066-Z, 1993.
Hanson, R. E., and Schweickert, R. A.: Chilling and brecciation of a Devonian rhyolite sill intruded into wet sediments,
northern Sierra Nevada, California. The Journal of Geology, 90(6), 717-724, https://doi.org/10.1086/628726, 1982.
Hildreth, W., Fierstein, J., and Calvert, A.: Early postcaldera rhyolite and structural resurgence at Long Valley Caldera,
California, J. Volcanol. Geoth. Res., 335, 1–34, https://doi.org/10.1016/j.jvolgeores.2017.01.005, 2017.
Jentsch, A., Jolie, E., Jones, D.G., Curran, H.T., Peiffer, L., Zimmer, M., and Lister, B.: Magmatic volatiles to assess
permeable volcano-tectonic structures in the Los Humeros geothermal field, Mexico. J. Volcanol. Geoth. Res., 394,
106820 https://doi.org/10.1016/j.jvolgeores.2020.106820, 2020.
Juárez-Arriaga, E., Böhnel, H., Carrasco-Núñez, G., and Mahgoub, A. N.: Paleomagnetism of Holocene lava flows from
Los Humeros caldera, eastern Mexico: Discrimination of volcanic eruptions and their age dating. Journal of South
American Earth Sciences, 88, 736-748, https://doi.org/10.1016/j.jsames.2018.10.008, 2018.





Le Maitre, R.W., Streckeisen, A., Zanettin, B., Le Bas, M. J., Bonin, B., Bateman, P., Bellieni, G., Dudek, A., Efremova,
S., Keller, J., Lameyre, J., Sabine, P. A., Schmid, R., Sqrensen, H., and Woolley, A. R.: Igneous Rocks. A Classification
and Glossary of terms. Recommendations of the IUGS Subcommission on the Systematics of Igneous Rocks, Cambridge
University Press, 236 pp., 2002.
Lipman, P.W., Moore, J.G., and Swanson, D.A.: Bulging of the northern flank before the May 18 eruption: geodetic data.
U.S. Geol. Surv. Prof. Pap., 1250:143–156, 1981.
Lipman, P.W., Zimmerer, M.J., and McIntosh, W.C.: An ignimbrite caldera from the bottom up: Exhumed floor and fill
of the resurgent Bonanza caldera, Southern Rocky Mountain volcanic field, Colorado: Geosphere, 11, 1902–1947,
https://doi.org/10.1130/GES01184.1, 2015.
Lorenzo-Pulido, C.D.: Borehole geophysics and geology of well H43, Los Humeros geothermal field, Puebla, Mexico,
United Nations University,Geothermal Training Programme Report, 23, 387-425, 2008.
Lucci, F., Carrasco-Núñez, G., Rossetti, F., Theye, T., White, J. C., Urbani, S., Azizi, H., Asahara, Y., and Giordano, G.:
Anatomy of the magmatic plumbing system of Los Humeros Caldera (Mexico): implications for geothermal systems,
Solid Earth, 11, 125–159, https://doi.org/10.5194/se-11-125-2020, 2020.
Martínez-Serrano, R. G.: Chemical variations in hydrothermal minerals of the Los Humeros geothermal system, Mexico.
Geothermics, 31(5), 579-612, https://doi.org/10.1016/S0375-6505(02)00015-9, 2002.
Mattsson, T., Burchardt, S., Almqvist, B.S.G, and Ronchin, E.: Syn-Emplacement Fracturing in the Sandfell Laccolith,
Eastern Iceland—Implications for Rhyolite Intrusion Growth and Volcanic Hazards. Front. Earth Sci. 6:5.
https://doi.org/10.3389/feart.2018.00005, 2018.
McConnell, V. S., Shearer, C. K., Eichelberger, J. C., Keskinen, M. J., Layer, P. W., and Papike, J. J.: Rhyolite intrusions
in the intracaldera Bishop tuff, Long Valley caldera, California. J. Volcanol. Geoth. Res., 67(1-3), 41-60,
https://doi.org/10.1016/0377-0273(94)00099-3, 1995.
Montanari, D., Bonini, M., Corti, G., Agostini, A., and Del Ventisette, C.: Forced folding above shallow magma
intrusions: Insights on supercritical fluid flow from analogue modelling, J. Volcanol. Geoth. Res., 345, 67–80,
http://dx.doi.org/10.1016/j.jvolgeores.2017.07.022, 2017.
Norini, G., Groppelli, G., Sulpizio, R., Carrasco-Núñez, G., Dávila-Harris, P., Pellicioli, C., Zucca, F., and De Franco,
R.: Structural analysis and thermal remote sensing of the Los Humeros Volcanic Complex: Implications for volcano
structure and geothermal exploration, J. Volcanol. Geoth. Res., 301, 221–237,
https://doi.org/10.1016/j.jvolgeores.2015.05.014, 2015.
Norini, G., Carrasco-Núñez, G., Corbo-Camargo, F., Lermo, J., Hernández Rojas, J., Castro, C., Bonini, M., Montanari,
D., Corti, G., Moratti, G., Chavez, G., Ramirez, M., and Cedillo, F.: The structural architecture of the Los Humeros
volcanic complex and geothermal field, J. Volcanol. Geoth. Res., 381, 312–329,
https://doi.org/10.1016/j.jvolgeores.2019.06.010, 2019.
Norini, G. and Groppelli, G.: Comment on "Estimating the depth and evolution of intrusions at resurgent calderas: Los
Humeros (Mexico)" by Urbani et al. (2020), Solid Earth, 11, 2549–2556, https://doi.org/10.5194/se-11-2549-2020, 2020.
Olvera-García, E., Bianco, C., Víctor Hugo, G. M., Brogi, A., Liotta, D., Wheeler, W., ... and Ruggieri, G.: Geology of
Las Minas: an example of an exhumed geothermal system (Eastern Trans-Mexican Volcanic Belt). J. Maps, *16*(2), 918-
926, https://doi.org/10.1080/17445647.2020.1842815, 2020.
Peiffer, L., Carrasco-Núñez, G., Mazot, A., Villanueva-Estrada, R., Inguaggiato, C., Romero, RB, Rocha Miller, R.,
Hernández Rojas, J.: Soil degassing at the Los Humeros geothermal field (Mexico). Journal of Volcanology and
Geothermal Research, 356, 163-174. https://doi.org/10.1016/j.jvolgeores.2018.03.001, 2018.



Petronis, M.S, van Wyk de Vries, B. and Garza, D.: The leaning Puy de Dôme (Auvergne, France) tilted by shallow
intrusions. Volcanica, 2, 161-189. https://doi.org/10.30909/vol.02.02.161186, 2019.
Philpotts, A., and Ague, J.: Principles of igneous and metamorphic petrology. Cambridge University Press, 667 pp., 2009.
Poppe S., Holohan E.P., Galland O., Buls N., Van Gompel G., Keelson B., Tournigand P.-Y., Brancart J., Hollis D., Nila
A. and Kervyn M.: An Inside Perspective on Magma Intrusion: Quantifying 3D Displacement and Strain in Laboratory
Experiments by Dynamic X-Ray Computed Tomography. Front. Earth Sci. 7:62.
https://doi.org/10.3389/feart.2019.00062, 2019.
Prol-Ledesma, R.M.:. Reporte de los estudios petrograficos y de inclusiones fluidas en nucleos de pozos de exploracion
en el campo geotermico de Los Humeros, Puebla, Mexico. Comunicaciones Tecnicas, Instituto de Geofisica, UNAM:
(86), pp. 1–75, 1988.
Prol-Ledesma, R.M.: Pre- and post-exploitation variations in hydrothermal activity in Los Humeros geothermal field,
Mexico. J. Volcanol. Geoth. Res., 83, 313–333, https://doi.org/10.1016/S0377-0273(98)00024-9, 1998.
Rabiee, A., Rossetti, F., Asahara, Y., Azizi, H., Lucci, F., Lustrino, M., and Nozaem, R.: Long-lived, Eocene-Miocene
stationary magmatism in NW Iran along a transform plate boundary. Gondwana Res., 85, 237-262,
https://doi.org/10.1016/j.gr.2020.03.014, 2020.
Rojas-Ortega, E.: Litoestratigrafia, petrografia y geoquımica de la toba Llano, y su relacion con el crater el Xalapazco,
Caldera de Los Humeros, Puebla, MS thesis, pp. 129, San Luis Potosı, Mexico: IPICYT,
https://colecciondigital.cemiegeo.org/xmlui/handle/123456789/509, 2016.
Rossetti, F., Aldega, L., Tecce, F., Balsamo, F., Billi, A., and Brilli, M.: Fluid flow within the damage zone of the
Boccheggiano extensional fault (Larderello–Travale geothermal field, central Italy): structures, alteration and
implications for hydrothermal mineralization in extensional settings. Geological Magazine, 148(4), 558-579,
https://doi.org/10.1017/S001675681000097X, 2011.
Sparks, R. S. J., Annen, C., Blundy, J. D., Cashman, K. V., Rust, A. C., and Jackson, M. D.: Formation and dynamics of
magma reservoirs. Philos. T. Roy. Soc. A, 377(2139), 20180019, https://doi.org/10.1098/rsta.2018.0019, 2019.
Torres-Rodriguez, M.A.: Characterization of the Reservoir of the Los Humeros, México, Geothermal Field, Proceedings
of the World Geothermal Congress 1995, Florence, Italy, May 18-31, vol. 3, pp. 1561-1567, 1995.
Urbani, S., Giordano, G., Lucci, F., Rossetti, F., Acocella, V., and Carrasco-Núñez, G.: Estimating the depth and evolution
of intrusions at resurgent calderas: Los Humeros (Mexico). Solid Earth, 11(2), 527-545, https://doi.org/10.5194/se-11-

508 527-2020, 2020.

Van Loon, A. J., and Wiggers, A. J.: Holocene lagoonal silts (formerly called "sloef") from the Zuiderzee. Sediment.
Geol., 13(1), 47-55, https://doi.org/10.1016/0037-0738(75)90049-4, 1975.
Van Loon, A. J., and Wiggers, A. J.: Metasedimentary "graben" and associated structures in the lagoonal Almere Member
(Groningen Formation, The Netherlands). Sediment. Geol., 16(4), 237-254, https://doi.org/10.1016/0037-

513 0738(76)90001-4, 1976.

Verma, S. P.: Heat source in Los Humeros geothermal area, Puebla, Mexico, Geoth. Res. T., 9, 521–525, 1985.
Yang, R., and van Loon, A. T.: Early Cretaceous slumps and turbidites with peculiar soft-sediment deformation structures
on Lingshan Island (Qingdao, China) indicating a tensional tectonic regime. J. Asian Earth Sci., 129, 206-219,
https://doi.org/10.1016/j.jseaes.2016.08.014, 2016.
Wernicke, B., and Burchfiel, B. C.: Modes of extensional tectonics. J. Struct. Geol., 4(2), 105-115, 1982.





Figure 1: a) Panoramic view showing the top of the draping unconformity surface (green dashed line) of the Cuicuiltic Member fall deposit covering the Las Papas scarp. b) Outcrop image along the Las Papas scarp showing the unaltered and underformed



Cuicuiltic Member uncomfomably lying on the Xoxotic Tuff.  c) Outcrop scale image of the LH26-1a site, investigated by both
Urbani et al. (2020) and Lucci et al. (2020), showing an altered trachyandesite lava covered by unaltered Cuicuiltic Member
layers along the Maxtaloya scarp close to the H6 well. Intraformational penecontemporaneous small-scale faults are visible in
upper layers of the Cuicuiltic Member deposit. d-g) Hydrothermal alteration associated with normal faults and joints within
the apical depression of the Loma Blanca bulge. f) NNE-SSW-striking Loma Blanca main fault showing reddish alteration on
its plane. g) Detail of the reddish hydrothermal alteration. h-i) Outcrop images of the active solfatara located 300 m away from
the H4 well, at the southern termination of the Loma Blanca fault, showing hydrothermal alteration  of both post-caldera
trachyandesites and overlying Cuicuiltic Member fall deposit.




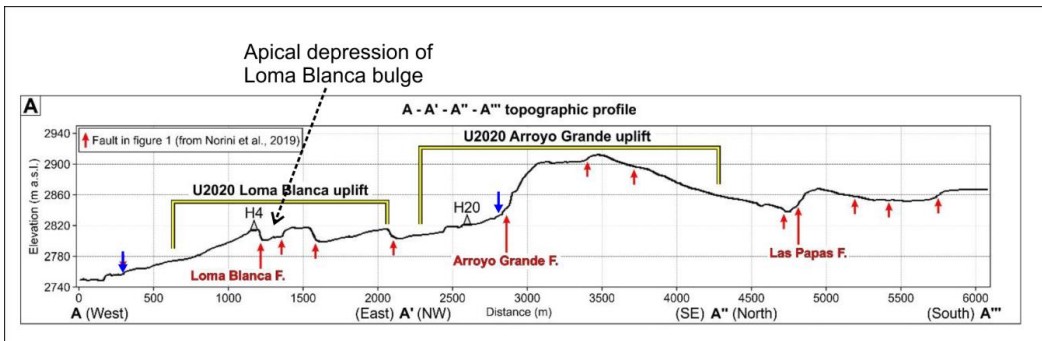

**Figure 2: Trace of the A-A'-A''-A''' topographic profile of N&G2020 showing the apical depression of the Loma Blanca bulge and reverse faults (blue arrows) at the base of the Arroyo Grande and Loma Blanca bulges identified by (Norini et al., 2019). Modified from Fig. 4a of N&G2020.**




**Figure 3: Lithostratigraphic columns of the wells a) H10, b) H20, c) H25, d) H26 and e) H42 as proposed by Carrasco-Núñez**
**et al. (2017b; CN17 in figure), Urbani et al. (2020; U20 in figure) and Cavazos-Alvarez et al. (2020; CA20 in figure). Felsic or**
**rhyolitic bodies within the volcanic sequence are indicated by red ellipses, whereas the newly identified andesitic lithic-breccias**
**within the intracaldera Xaltipan Ignimbrite deposits (Cavazos-Alvarez et al., 2020) are indicated by blue ellipses.**




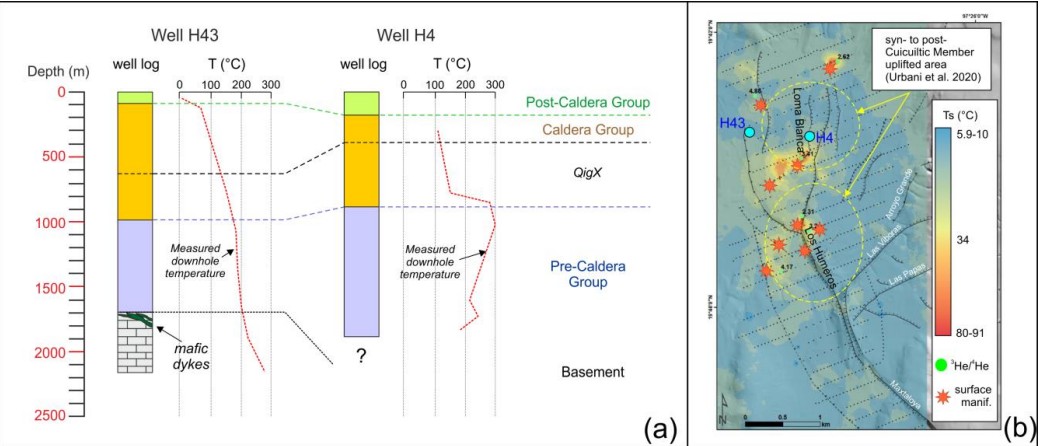

**Figure 4: a) In-depth correlation of lithostratigraphic units for H4 and H43 geothermal wells (after Areallano et al., 2003; Carrasco-Núñez et al., 2017b; Urbani et al., 2020). Measured downhole temperature profiles for well H4 (Torres-Rodriguez, 1995; Prol-Ledesma, 1988, 1998; Martinez-Serrano, 2002) and well H43 (Lorenzo-Pulido, 2008) are reported. b) Interpolation map of soil temperatures measured at Los Potreros Caldera (modified after Jentsch et al., 2020; GEMex, 2019a). Orange stars showing locations of hydrothermal surface manifestations are after Jentsch et al. (2020). Geothermal wells H4 and H43 are also reported. Yellow dashed ellipses indicate the syn- to post-Cuicuiltic Member eruption uplifted area as proposed by Urbani et al. (2020).**



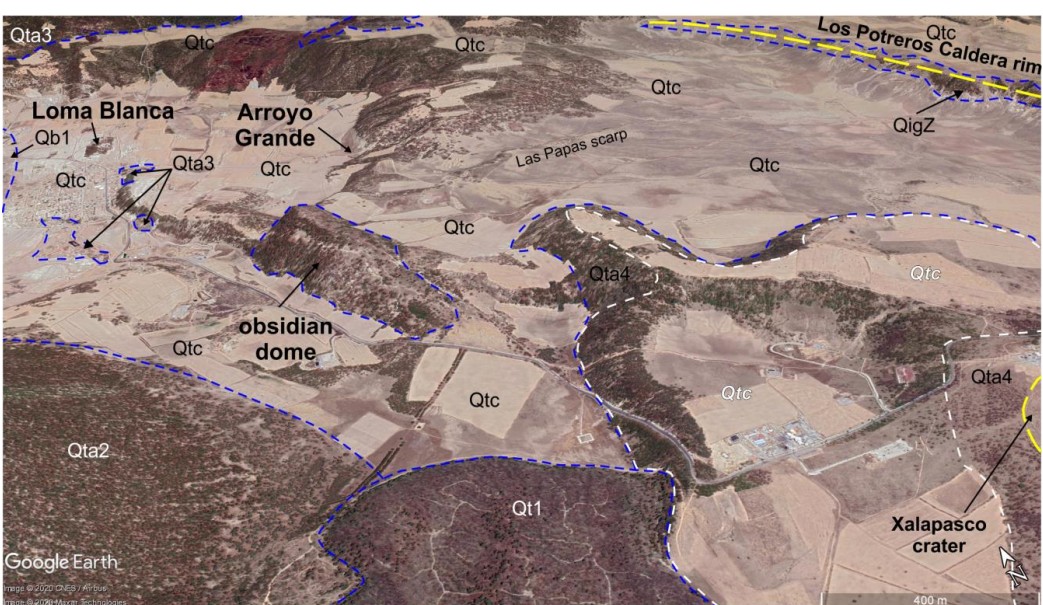

556

**Figure 5: Perspective view from a satellite image of the Los Potreros Caldera floor (Image Landsat from Google Earth Pro,**

**2020, Inegi-Maxar Technologies; courtesy of Google). The dashed blue lines outline the lava domes and flows (Qta2, Qta3, Qb1,**

**Qta4, Qt1) mapped by Carrasco-Núñez et al. (2017a) whereas the dashed white lines outline the mapping of the Cuicuiltic**

**Member (Qtc) from Urbani et al. (2020).**
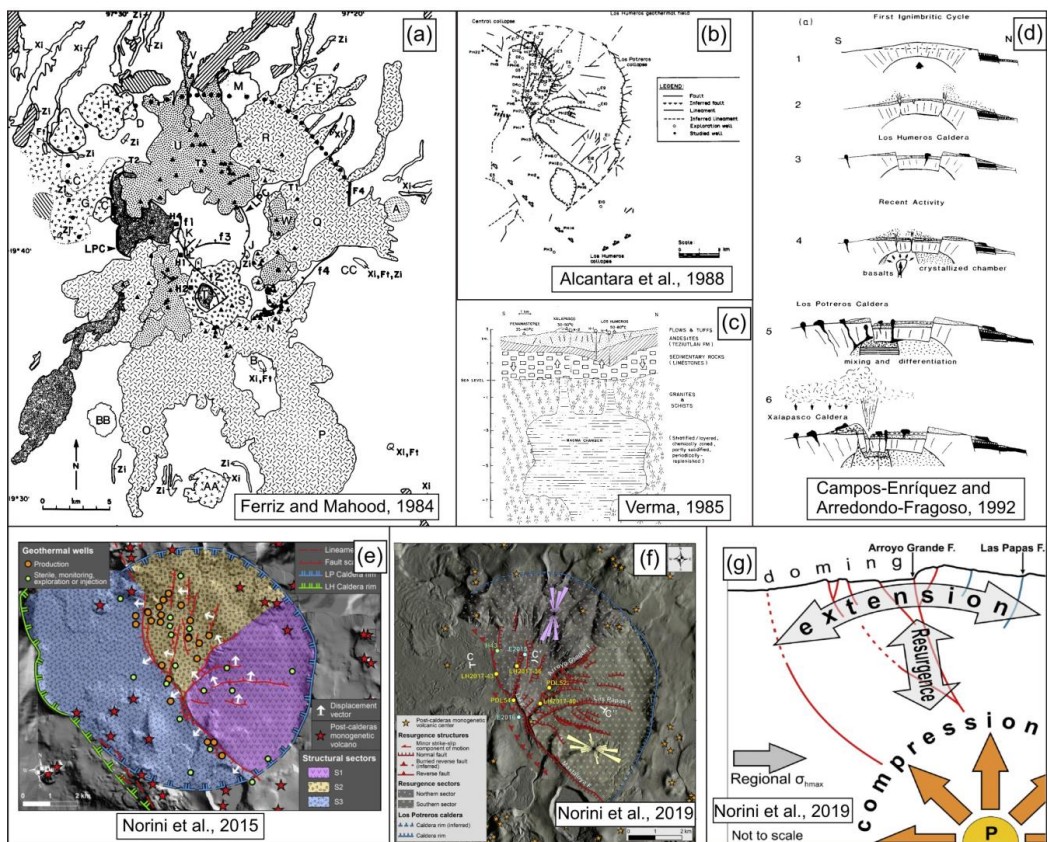

Figure 6: a) The simplified geological map of the Los Humeros volcanic center as proposed by Ferriz and Mahood (1984). b) Schematic map of the Los Potreros caldera showing the main structures and the exploration wells drilled before the 1988. This map was presented by Alcantara et al. (1988) based on unpublished map by CFE. c) Conceptual model of the single voluminous magma chamber underlying the Los Humeros volcanic center as proposed by Verma (1985). d) Schematic representation of the evolution of Los Humeros volcanic complex by Campos-Enriquez and Arredondo-Fragoso (1992) where magmatism, eruptive styles, inflation and deflation phenomena are all correlated to the activity of the single voluminous and shallow-seated magma chamber of Verma (1985). e) Morphostructural map of the Los Potreros caldera with interpretation of the sectorial resurgence as proposed by Norini et al. (2015). f) Morphostructural map of the Los Potreros caldera with interpretation of the sectorial resurgence as proposed by Norini et al. (2019). g) Schematic not to scale structural interpretation of the post-caldera resurgence at Los Humeros induced by a unique pressure source at depth as proposed by Norini et al. (2019).





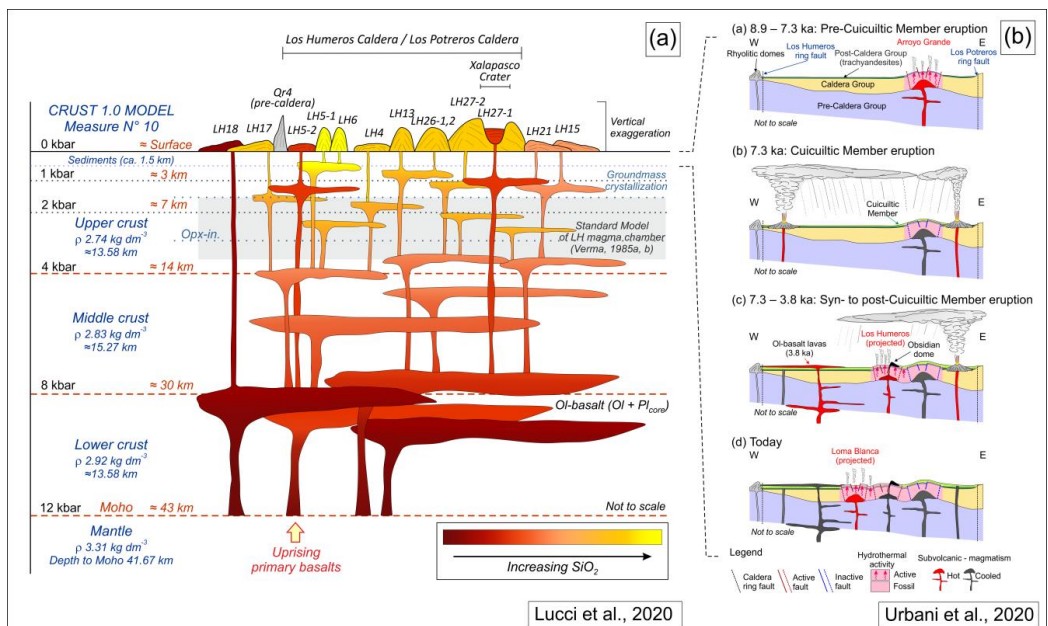

572

**Figure 7: a) Schematic representation (not to scale), by Lucci et al. (2020), of the magmatic plumbing system feeding the Los Humeros post-caldera stage activity, beneath the Los Humeros Caldera, as derived by pressure-temperature estimates obtained from mineral-liquid thermobarometry models. The model is integrated with the crustal structure (see Lucci et al., 2020, for further explanations). b) Schematic model, by Urbani et al. (2020), of the evolution and of the subsurface structure of the Los Potreros caldera floor. Multiple magmatic intrusions located at relatively shallow depth (< 1km) are responsible for the localized bulging of the caldera floor (Arroyo Grande, Los Humeros, and Loma Blanca uplifted areas). The Cuicuiltic Member eruption is assumed as a time-marker in the evolution of the intracaldera domain.**

580