# Peer review of "Structural studies in active caldera geothermal systems. Reply to"

_Solid Earth, 2020_

## Referee Comment (RC2)

[referee-annotated manuscript omitted]

---

## Author Response (AR2)

**Reply to anonymous reviewer 1**

We thank the reviewer for the comments provided on our Reply. In the following, we detail our response to the reviewer's remarks.

COMMENT: The reviewer agrees in principle with the arguments developed by Urbani et al in this manuscript in response to the points made by Norni and Groppelli (2020). The counterarguments made in this manuscript to the points made by Norni and Groppelli (2020) are reasonable and provide a clearer view of the authors' claims.

ANSWER: We thank the reviewer for this comment.

COMMENT: However, some of the discussing points made by Urbani et al and Norni and Groppelli are derived from different observations and data set, Therefore, a more comprehensive examination, including field research, is needed to determine which claim is closer to the truth, which is beyond the scope of this peer review. As pointed by the authors, the further investigations in the Los Humeros caldera are necessary to confirm its structural evolution.

ANSWER: We fully agree as pointed out in our Reply.

COMMENT: Generally, formation of many shallow intrusions during "post-caldera stage" modifies the original structures and hides the signals from deep magma system. As develop the shallow intrusion system, investigation of the bottom structure of caldera (magma chamber) becomes difficult from the approach of structural geology from the ground surface. Therefore, the discussion of the presence/absence of magma chamber beneath a caldera system should be careful and should be based on the direct geophysical investigations (seismic, magnetotelluric, and gravity), not only from geological evidences.

ANSWER: We agree.

COMMENT Lines 50-52: Basically I support this selection to highlight the first-order structure. To show the validity of this method, can you show the percentage that the sum of the displacements of the selected faults represents of the total displacements in a given section? This is to show that the contribution of the unselected faults is not important for the overall deformation structure.

ANSWER: As discussed in the Reply, we have measured fault strands/systems that either individually or summed total in m-scale displacements or more. Minor faults typically show offsets in the order of cm- to dm-scale and in many instances are intraformational. First and second order structures usually do not occur at same localities, however where this happens, the impact of these minor faults at any given section on the total displacement is one to two order of magnitude less

respect to first order ones. The only way to better quantify the relative contribution to the total displacement is the measure of scan lines along a number of representative outcrops for each structure, which was not in the scope of our work.

COMMENT L58-63: I agree that the outcrop on the Las Papas scarp shown in Fig. 1b shows no clear evidence of faulting. However, in general, we should recognize that the topographic displacement by faulting can be modified by the erosion and the location of present "topographic step" does not coincide the place of the underground structural fault. The reviewer is not sure if the location of this studied outcrop (shown in Fig. 2) is a critical point for assessing the activity of Las Papas scarp. Anyway, if the structure is active, the place of structure is almost coinciding with the surface topographic relief.

ANSWER: We agree with the reviewer.

COMMENT Lines 75-76; If these minor faults are non-tectonic and formed by the gravitational instability, their orientation and kinematics should be associating the local topography. Can you show it?

ANSWER: The Reviewer refers to faults presented by Norini and Groppelli (2020) in their Figure 2. Therefore, we cannot add/show more than what we see in this figure. In general, however, faults of any kind likely bear the same trend of the associated topography.

COMMENT Lines 96-98: Here, you should put a connecting logic between the activity of faults and the creation and maintenance of the pathway of hydrothermal fluid as not all geothermal field distribute along "active" fault, in general.

ANSWER: We agree with Rev1 that the sentence should be rephrased. Given the subject of the Reply we were focusing only on fracture-controlled fluid flow but it is correct to say that geothermal fields do develop also associated with primary porosity. We will add this information in the revised version.

COMMENT Lines 137-141: Formation of "visible" apical depression at the top of uplift may be also controlled by many factors such as the intensity of horizontal extension at the top of bulge and mechanical properties of the materials at the bulge. You can put the comparison of the difference of these factors between the field and model, to show the reason why the model shows an apical depression whereas the fields not always.

ANSWER: The setting and scaling of the analogue models in Urbani et al (2020) was designed to test and expand the validity of the relationships between surface deformation and intrusion depth(s) for sub-circular sources. To this purpose, the experiments were designed to ensure the formation of an apical depression by pushing the silicone to the surface. Other kinds of experiments should be carried out to compare one-to-one nature and models, but this was not in the scope of our work.

COMMENT Lines 152-154: Can you show the orientation and kinematic of these reverse faults shown in Figure 2, to show the relationship between the formation of these bulges and reverse faults?

ANSWER: As clearly stated in the Reply and caption of Fig.2, the indicated reverse faults are redrawn from Norini and Groppelli (their Figure 4a), not our dataset, so we cannot show more than what shown in Fig.2.

COMMENT Lines 165-168: Though I strongly agree the difficulty of the identification of shallow intrusion from subaerial lava only from well log, can you find any supporting description for intrusion, not normal lava flow, such as contact metamorphism and chilling texture at the upper boundary of these potential intrusion? If yes, it is a direct evidence for the presence of shallow intrusions.

ANSWER: Chilled margins are described in a number of these rhyolitic bodies, but this evidence was not yet published. Therefore, we cannot add any reference. We agree with reviewer that these structures can be interpreted as evidence of their intrusive nature.

COMMENT Lines 202-206 and Figure4(a): Though the temperature profile of Well H4 suggests the presence of heat source at around ~1000m depth, if you combine it with the stratigraphic description and interpretation of the well, you can show more strongly the possibility. Because the depth of temperature maximum in the well H4 is close to the lithostratigraphic boundary between pre-caldera group and QigX, the temperature profile may reflect the distribution of hydrothermal fluid at the boundary. Can you show the presence of potential intrusion on the lithostratigraphic column (like Figure 3)?

ANSWER: More detailed lithostratigraphic columns for well H4 have not been yet published. However, real intrusions may have variable shapes and emplacement depths that, if close enough, may affect the thermal profile of a single well, even if they are not directly drilled. In any case, we take this opportunity to clarify that the study of Urbani et al (2020) was not designed to deterministically model individual shallow intrusions, but to show that observables such as local bulging, deformation, faulting and alteration patterns can be reconciled with a scenario of shallow emplacement of acidic magma bodies. More deterministic assessments were beyond the scope of

the work and possibly need other kinds of investigation methods, such as high resolution active seismic profiles.

COMMENT Lines 216-219: I guess that the location of the dated sample HK-14-08 (U-Th age 44.8+-1.7ka) reported by Carrasco-Nunez et al.2018 corresponds the "Obsidian dome" indicate at the center of Figure 5. If so, the age of this dome is 44 ka, much older the Cuicuiltic member erupted 7.3 ka. If so, this "dome" is the uplifted block by faults. Instead, Qt1 and Qta2 are clearly younger than Cuicuiltic member, judging from the distribution shown in Figure 5 and ages reported by Carrasco-Nunez et al.2018. The eruption sources of these lavas are very close to the Maxtaloya-Los Humeros faults.

ANSWER: As documented in the Reply, the obsidian dome is largely uncovered by the Cuicuiltic member. The location of the sample HK-14-08 is at the base of the dome and U-Th ages represent crystallization ages of zircons in the magma chamber (Carrasco Nunez et al 2018). The dome is likely polyphase, with an earlier phase of emplacement older than the Cuicuiltic and a later one younger. We agree with the Reviewer that the dome could have been affected by exhumation after 7.3 ka. The emplacement/exhumation of the obsidian dome together with the nearby faulting of the Cuicuiltic member by tens of meters of displacement at the site of the Los Humeros fault indicate that this fault strand was active later than 7.3 ka. By contrast, the fault displacement drastically reduces southward along the Maxtaloya fault. This evidence supports our interpretation of the Maxtaloya-Los Humeros faults as segmented fault strands, with diachronous activity during the Holocene.

Yours sincerely,

Stefano Urbani, Guido Giordano, Federico Lucci, Federico Rossetti and Gerardo Carrasco Nuñez.

**Reply to reviewer John Browning**

We warmly thank the reviewer John Browning for the comments provided on our Reply. We will change the text according to his suggestions. Hereafter, we also provide a detailed reply on the reviewer's major comments:

COMMENT Line 34: Are they referring to the quality of the data? I would suggest the word quality is added after the word poor. Or are they referring to the coverage or absence of data? Either way i think there needs to be a qualifying word after the word poor to focus what is 'poor' about the data.

ANSWER: We suppose that they are referring to the absence of data. Therefore, we will change the word poor with "lack of".

COMMENT Line 96: Only for interest the authors may also wish to read the following manuscript which reported similar results but for the Karliova triple junction region in Turkey. I am in no way suggesting this should be cited here, i just mention it for interest. Karaoğlu, Ö., Bazargan, M., Baba, A. and Browning, J., 2019. Thermal fluid circulation around the Karliova triple junction: Geochemical features and volcano-tectonic implications (Eastern Turkey). Geothermics, 81, pp.168-184.

ANSWER: We thank the reviewer for bringing this interesting article to our attention. We see no harm in citing the suggested article that can be of interest also for the external reader. Therefore, on our own initiative, we will cite it in the revised version of the manuscript.

COMMENT Line 154: I think that this final statement is also slightly unclear. What do you mean by 'their own statements are really unclear'? This should be reworded to be stronger and more precise. Do you mean that their comments are contradictory?

ANSWER: We will replace the term "unclear" with "contradictory" in the revised text.

COMMENT Lines 157-158: I find this sentence hard to follow and difficult to understand what you want to convey. Please rephrase.

ANSWER: We will rephrase the sentence as follows:"Norini and Groppelli 2020 invoke the thermal profile and the stratigraphy of just one well log (H4 well, drilled on the top of the Loma Blanca bulge) to claim the lack of validation of the models proposed in Urbani et al. (2020)"

COMMENT Line 301: I find this statement somewhat antagonizing. I would suggest that the authors remove this to avoid any further back and forth comments which at this stage should really be avoided.

ANSWER: We will delete the statement as suggested by the reviewer.

COMMENT Lines 303-308: I find all of these final comments somewhat strange and i would suggest they be removed. I agree in part that much of this comment stream seems unnnessary, especially since many of the authors of the competing comments have worked and published together on the same topic. So there appears to be al ot of cross-over which is verging on innappropriate.

ANSWER: We totally agree with the reviewer that the back and forth comments made by the authors of the "comment" are inappropriate as already expressed by one of the co-authors in the discussion forum. We think that these final statements are not against the authors of the "comment" and were added to the text to explain the context from which such scientific debate has started. This would have helped to clarify the nature of part of the comment stream that we believe inappropriate and unnecessary as already pointed out by the reviewer. However, we recognize that such a matter

might not be of interest for an external reader thus, if the Editor agrees, we will remove it from the text as suggested.

Yours sincerely,

Stefano Urbani, Guido Giordano, Federico Lucci, Federico Rossetti and Gerardo Carrasco Nuñez.